# Lactic Acid Bacteria Simultaneously Encapsulate Diverse Bioactive Compounds from a Fruit Extract and Enhance Thermal Stability

**DOI:** 10.3390/molecules27185821

**Published:** 2022-09-08

**Authors:** Fang Dou, Rewa Rai, Nitin Nitin

**Affiliations:** 1Department of Food Science and Technology, University of California, Davis, CA 95616, USA; 2Department of Biological and Agricultural Engineering, University of California, Davis, CA 95616, USA

**Keywords:** probiotic bacteria, polyphenols, encapsulation, antioxidant capacity, confocal imaging, HPLC-DAD

## Abstract

This study develops an innovative cell-based carrier to simultaneously encapsulate multiple phytochemicals from a complex plant source. Muscadine grapes (MG) juice prepared from fresh fruit was used as a model juice. After incubation with inactivated bacterial cells, 66.97% of the total anthocyanins, and 72.67% of the total antioxidant compounds were encapsulated in the cells from MG juice. Confocal images illustrated a uniform localization of the encapsulated material in the cells. The spectral emission scans indicated the presence of a diverse class of phenolic compounds, which was characterized using high-performance liquid chromatography (HPLC). Using HPLC, diverse phytochemical compound classes were analyzed, including flavanols, phenolic acid, hydroxycinnamic acid, flavonols, and polymeric polyphenols. The analysis validated that the cell carrier could encapsulate a complex profile of bioactive compounds from fruit juice, and the encapsulated content and efficiencies varied by the chemical class and compound. In addition, after the heat treatment at 90 °C for 60 min, >87% total antioxidant capacity and 90% anthocyanin content were recovered from the encapsulated MG. In summary, these results highlight the significant potential of a selected bacterial strain for simultaneous encapsulation of diverse phenolic compounds from fruit juice and improving their process stability.

## 1. Introduction

Plants produce a wide variety of human health-benefiting compounds [1]. Numerous studies have shown that plants contain rich and complex profiles of phytochemicals, including anthocyanins and other polyphenols [2,3]. The bioactive functionalities of these compounds, such as antioxidants, anti-inflammatory, anti-carcinogen, and maltase inhibition, may deter or prevent chronic diseases such as cancer [4], cardiovascular diseases [5], and diabetes [6]. Therefore, there is an increasing demand for the integration of these bioactive compounds with food. The challenges, however, lie in the complex chemical profile of the plant-based materials and the lack of stability under processing and storage conditions of bioactive compounds [7,8,9]. Several extraction techniques have been developed for separating these polyphenolic compounds from plant materials, such as absorption and ion-exchange technologies with microporous resins [10,11], liquid–liquid extraction [12], and membrane filtration [13]. Some of the key limitations of these approaches include inefficient extraction of a large diversity of polyphenolic compounds, labor extensive processes using high volumes of organic solvent, and limited protections for the sensitive compounds against degradation after extraction [10,14]. Thus, there is a significant need to develop environmentally and economically friendly approaches to efficiently separate and stabilize the high-value bioactive compounds from plant sources and deliver their health-promoting functionalities. Furthermore, these solutions may also need to address the sensory challenges with some of the plant bioactives.

Microencapsulation processes have been applied to concentrate, protect, and facilitate the incorporation of the polyphenolic compounds from plant extracts into food and pharmaceutical matrices [15,16]. By definition, microencapsulation refers to technologies of formulating solids, liquids, or gaseous materials into microparticles or dispersion, with diameters typically ranging between 0.1 and 1000 μm [17,18]. The industrial applications of the microencapsulation process offer a wide array of advantages in delivering polyphenolics, such as: (a) protecting encapsulated phyto-active compounds from degradation during processing and storage [8,19]; (b) controlling and targeting the release of the encapsulated compounds [18]; and (c) tailoring the undesirable physical characteristics of the polyphenolic compounds such as solubility, smell, and taste [20,21], etc. The most common coating materials are polymers, which include natural polymers (e.g., polysaccharides, proteins, and lipids [19]) and synthetic polymers (e.g., poly (lactic acid), poly (glycolic acid), and copolymers [22]). The shell for these microparticles is often formed using both physical and chemical processes such as spray drying and coacervation [15,19]. Encapsulation systems can provide protection for the bioactive compounds using a combination of exogenous preservatives and coating materials. However, conventional encapsulation systems lack mechanisms to selectively bind phytochemicals from plant extracts and to protect these health-promoting phytochemicals, often without exogenous preservatives [23,24].

Biological microscale structures, such as microbial cells, have emerged as promising encapsulation carriers for bioactive compounds. The results of these prior studies illustrate that microbial cells, such as yeast cells, can bind and encapsulate purified phytochemicals and protect them from oxidative and thermal stresses [25]. Thus, the key advantages of microbial cell-based encapsulation systems are (a) these pre-existing cell-based microstructures eliminate the need for expensive processes used for creating these microstructures from biopolymers; (b) eliminate the exposure of phytochemicals to heat, oxygen, and other physical factors that may deteriorate the encapsulated compounds, and (c) reduce/eliminate the need of exogenous preservatives and antioxidants [24,25,26]. The current studies using microbial cells have focused on purified plant-derived compounds, and to the best of our knowledge, no study has evaluated the role of microbial carriers for binding and encapsulation of diverse phytochemicals from plant juices or concentrations. Furthermore, most of the studies using microbial cells have focused on yeast cells as a model system with limited emphasis on bacterial cells for the binding and encapsulation of complex polyphenolic compounds. Lactic acid bacteria are widely recognized for their role in food fermentation and are increasingly evaluated for their probiotic functionality, thus having a significant potential to impact food systems and human health [27]. Despite this potential, there is limited evaluation of the potential of lactic acid bacteria to encapsulate diverse bioactives. In addition, the unique structural and compositional features of lactic acid bacteria, including cell wall biopolymer composition and structure [28], significantly high single-cell protein content [29], bio-affinity to interact with the gut [30], and high levels of antioxidant activity [31] due to various small molecules, peptides, proteins, or enzymes makes these bacterial cells a preferred encapsulation matrix for plant-based bioactives.

Thus, the focus of this study was to evaluate the binding and encapsulation of phytochemicals from a model fruit juice using inactivated probiotic bacteria *Lactobacillus casei.* Muscadine grapes (MG) juice was selected in this study since these are popular and highly valued fruits with rich phytochemical profiles and antioxidant properties [32,33]. Inactivated probiotic bacteria *L. casei* was selected as it is a widely used probiotic strain from a *Lactobacillus* family and thus is widely accepted as a beneficial ingredient in food systems [34,35]. Heat-inactivated cells were selected for the encapsulation to limit the metabolism of the encapsulated compounds as well as to increase the permeability of the cells for the fruit phytochemicals. Furthermore, inactivated probiotic cells retain some of the beneficial probiotic functions, as illustrated by recent studies [36,37]. To develop a simple approach that can be adapted by other researchers, a simple incubation process was utilized to bind and encapsulate phytochemicals from the juice matrix of MG using bacterial cells in this study. The anthocyanin content and antioxidant properties of the juice matrix before and after incubation was selected as an overall measure of encapsulation efficiency of complex phytochemicals from juice matrix. To further characterize the binding and localization of phytochemicals to bacterial cell matrix, multispectral fluorescence confocal imaging data was acquired. The binding and encapsulation yield of key selected phytochemical compounds from MG was also quantified using a high-performance liquid chromatography (HPLC). The effectiveness of the selected bacterial carrier to protect encapsulated phytochemical compounds was assessed based on the thermal treatment of both the encapsulated phytochemicals and the phytochemicals in a control juice matrix.

In summary, this study demonstrates the potential of using inactivated probiotic bacterial cell carrier for the binding and encapsulation of phytochemicals from a complex juice matrix and characterizes the binding and encapsulation efficiency of diverse phytochemicals and stability of encapsulated compounds in bacterial carriers using a combination of chemical analysis, spectral imaging, and antioxidant properties.

## 2. Results and Discussion

### 2.1. Total Antioxidant Capacity and Anthocyanin Content in Juice Matrices and Encapsulated Cell Carriers

Fruit juice contain a variety of phytochemicals, which in general can be classified into alkaloids, carotenoids, nitrogen-containing compounds, organosulfur compounds, and phenolics [38]. Many in vitro and in vivo studies support that the antioxidant property of the phytochemicals plays a major role in their essential health benefits such as anti-inflammation and anti-carcinogen [39,40]. To characterize the overall efficiency of encapsulating complex profiles of phytochemicals using the bacteria cell carriers, the relative encapsulation efficiency was measured based on the difference in antioxidant concentration of the juice sample before and after incubation with cells. To quantify this ratio, antioxidant concentration in juice sample and juice residue after the encapsulation process was measured using the FRAP (ferric reducing antioxidant power) assay [41]. The method for FRAP assay is described in detail in Section 3.6. These differences in the FRAP values before and after the encapsulation process reflect the relative amount of antioxidant compounds, including phenolics that are infused or bound to a selected cell-based micro-carrier. Table 1 shows the total antioxidant capacity of MG juice sample measured using the FRAP assay. The encapsulation efficiency in the selected bacterial carrier was 72.67% for MG. This percentage indicates the total fraction of antioxidant compounds bound and encapsulated in a bacterial cell carrier compared to the total antioxidant content in the juice sample. This result suggests that a simple incubation method allows phytochemicals to passively diffuse from a juice matrix to inactivated *L. casei* cells and results in an efficient binding and encapsulation of the antioxidant compounds in the cell carrier.

In addition to characterizing the encapsulated antioxidant content, the encapsulation efficiency of the anthocyanin pigments from the juice to the cells was also evaluated. Anthocyanin, being water soluble, is one of the major polyphenolic fractions in fruit juice and has a significant contribution to its antioxidant properties [42]. To assess the anthocyanin content in the juice before and after encapsulation, the juice matrix was extracted using methanol as described in the materials and methods section and the total anthocyanin content in the extract before and after incubation with cells was measured using a UV-Vis spectrophotometry. The measured absorbance at 530 nm was converted to an equivalent keracyanin chloride concentration (an anthocyanin standard) using a standard curve. Results show that MG juice had approximately 8.21 (±0.07) µM/mL of the equivalent keracyanin content. After incubation with inactivated bacterial cells, 66.97% of the total anthocyanin from the MG juice was encapsulated or bound to the cell carriers. In summary, these results highlight a significant potential of the selected bacterial strain for encapsulating antioxidants and anthocyanin family of compounds from a complex juice sample.

### 2.2. Confocal Images and Lambda Scans of Encapsulated Cells

To help visualize the encapsulated compounds and their intracellular distribution in the cell carriers, confocal multispectral fluorescence images were acquired based on the endogenous fluorescence signals of phytochemicals. The images were collected with a 405 nm excitation and an emission in the FITC channel from 500 to 550 nm. The fluorescence intensity of the cells in each image was quantified by randomly selecting 20 cells and measuring their mean pixel intensity using the ImageJ software. The mean background intensity was subtracted from the cell signals to remove the background signal.

As shown in Figure 1, the signal intensity of *L. casei* carriers increased approximately 24-fold upon incubation of cells with an MG juice (Figure 1b) as compared to the auto-fluorescence signal from the control cells (Figure 1a). Differences in the fluorescence signal intensity between the controls and the modified cells with juice phenolics was statistically significant with a *p*-value ≤ 0.05. The zoomed-in views in Figure 1b indicated that the cell carriers retained the cellular structure after the encapsulation process, and the encapsulated material was localized relatively uniformly across the intracellular compartment.

Further, lambda emission scans (Figure 2) were collected in the range of 470–670 nm with a 20 nm step size with an excitation wavelength at 405 nm. The results in Figure 2 revealed the fluorescence spectral profile of phenolic compounds of MG encapsulated in the *L. casei* cells. The cells encapsulated with MG juice showed a broad emission distribution over a range of 450–630 nm with a peak maximum around 515 nm and a secondary peak around 590 nm.

The broad emission range is usually associated with the presence of a diverse class of polyphenolic compounds. Based on the previous literature related to fluorescence properties of polyphenolics [43,44], the emission band between 533 nm and 595 nm mostly corresponds to anthocyanin content. MG spectra with the secondary emission around 590 nm indicates the presence of anthocyanin compounds in the cell carriers from juice matrix. In addition, the major peak in the MG spectra around 515 nm suggests a possible encapsulation of other phenolic compounds. Plant phenolics such as ferulic acid are known to have fluorescence emissions centered around 520 nm–530 nm [45,46,47].

The broadening of the peaks observed in Figure 2 could be attributed to other photoactive compounds present in the complex juice matrix. The shift in the emission range compared to the peaks observed from prior literature could also be caused by multiple factors. Anthocyanin polymerization during the juice processing and storage process could cause the emission to shift towards shorter wavelengths [48]. In addition, fluorescence emission spectrums are known to be sensitive to environmental factors, including the excitation wavelength, medium pH and polarity, present macromolecules, etc. [49]. Thus, it may contribute to the shifts observed in Figure 2. In this measurement, phenolic compounds that emit blue fluorescence (400 nm–470 nm) were not captured in Figure 2 due to the limitation of the available wavelength range in this imaging system. To address these gaps in the compositional analysis of encapsulated compounds, analytical measurements using a HPLC method with known standards were conducted.

### 2.3. Phenolic Profile of the Juice Matrix and Encapsulated Cell Carrier

Among the diverse groups of bioactive compounds present in the fruit and fruit juices, phenolic compounds constitute one of the largest and most diverse groups of phytochemicals [50]. To characterize the phenolic compounds profile of the juice and the encapsulation efficiency, the MG juice before and after incubation with cells was analyzed using HPLC and, based on these measurements, the encapsulation efficiency of the selected polyphenolics was quantified. The protocols for the evaluation of phenolics in a grape juice matrix were already developed by Oberholster et al. [51]. Target compound classes included in this study were flavanols (catechin and epicatechin, 280 nm), phenolic acid (gallic acid, 280 nm, caffeic acid, and coutaric acid, 320 nm), flavonols (quercetin and glycosylated myricetin, 360 nm), and polymeric polyphenols (280 nm). Catechin, epicatechin, gallic acid, and polymeric phenols were quantified using chromatograms at signal 280 nm, caffeic acid at 320 nm, and quercitin and glycosylated myricetin at 360 nm. These polyphenolic compounds have been determined to be among the leading polyphenolic compounds in a grape juice. The chromatograms of 20% MG juice matrix at signals 280, 320, and 360 nm before and after encapsulation are shown in Figure 3. Peaks corresponding to each analyzed phenolic compounds were assigned in chromatograms collected at each signal.

As observed from Table 2 and Figure 3, most of the investigated compounds were present in MG juice at different concentrations and had different levels of encapsulation efficiency. For flavanols, catechin and epicatechin were both present in the MG juice matrix. The catechin (16.78 mg/mL) concentration is significantly higher than epicatechin (3.18 mg/mL). Despite these differences in absolute concentration levels, 17.40% of catechin and 18.77% of epicatechin were encapsulated upon incubation of cells with MG juice.

For the phenolic acids, the concentration of gallic acid in MG juice was 1.69 mg/mL and its encapsulation efficiency was 18.43% in *L. casei* cells. The content and encapsulation efficiency for gallic acid was significantly higher than the amount of coutaric acid (0.07 mg/mL) and its encapsulation efficiency (10.24%) in *L. casei* cells. The amount of caffeic acid was below the detection limit in MG juice.

Among flavonols, 20% MG juice contains 21.70 mg/mL of quercetin. The encapsulation efficiency (2.83%) of quercetin in *L. casei* was limited as compared to the quantified flavanols and phenolic acids. Glucoside derivatives are commonly found in grapes and wines, particularly delphinidin-3-glucoside, petunidin-3-glucoside, and malvidin-3-glucoside (Revilla, 1999). MG juice contains 3.53 mg/mL myricetin 3-glycoside and its encapsulation efficiency was 69.85% upon incubation of cells with MG juice.

Another common abundant polyphenolic compound in MG juice was polymeric phenols. The 20% juice contains 26.09 mg/mL of polymeric phenols. The polymeric phenols identified using this protocol represent a mixture of polymeric pigments, which are formed based on reactions between grape anthocyanins and other components in the juice such as tannin, catechins, and proanthocyanidins [52,53]. A total of 97.97% of the polymeric phenol was infused into the cell carriers upon incubation with MG juice.

Taken together, the imaging and HPLC measurement results illustrate that cell carriers can simultaneously encapsulate diversity of bioactive compounds from a complex juice matrix. Compared to previous studies that have predominantly focused on yeast cells for the encapsulation of purified hydrophobic polyphenolic compounds [54,55], the results of this study suggest a potential of diverse cell carriers, including bacterial cell carriers, to simultaneously encapsulate multiple compounds from mixtures. Furthermore, since the encapsulation process was conducted using water soluble compounds in fruit juice, this study demonstrates that bacterial cell carriers can bind and encapsulate compounds from water extracts and juices. Together with prior studies, the results of this study illustrate the potential of cell carriers to encapsulate both hydrophobic and hydrophilic bioactives. The encapsulation process of these compounds from cell carriers can be attributed to both composition and structure of cell carriers. Besides the structural integrity that withstood the encapsulation process as shown in Figure 1, bacterial and yeast cell carriers have a relatively high fraction of protein content on a dry basis. In the case of *L. casei* cells, the protein content can be as high as 80% or higher on a dry basis. Similarly, the protein content in yeast cells can range from 25 to 60% on a dry basis [56,57]. In addition, cell carriers also express both soluble and structural proteins including membrane associated proteins. In previous studies, protein–polyphenolic interactions have been explored and the binding between protein isolates and polyphenolic compounds from juice or other plant extracts has been demonstrated [58,59,60]. Thus, it is likely that a relatively high concentration and diversity of proteins in micro-scale cells carriers significantly promote the binding of diverse polyphenols from a juice matrix. In addition to proteins, bacteria and yeast cells also contain a diversity of carbohydrate biopolymers mostly concentrated in cell walls and lipids that are integral parts of the cell membranes. Prior studies have shown interactions between polyphenols and cellular polysaccharides [61,62], and the binding mechanism could be attributed to a range of physical and chemical interactions [63]. The complex and porous structures and surface properties of the cell wall has also been proposed to be important for the binding process [63,64]. These compositions and cellular structures can provide a rich environment for the partitioning and compartmentalization of diverse compounds in cell-based carriers.

The results illustrate that the encapsulated content and efficiencies varied by the chemical class, compounds, and juice matrix. Quercetin as a monomeric flavonol showed low incorporation rates from MG juice, while the glycosylated myricetin has a significantly higher encapsulation efficiency (69.5% from MG juice). In contrast, polymeric polyphenols yielded the highest encapsulation efficiency among all compounds tested from MG juice (at approximately 97%). This trend of differences in encapsulation efficiency of compounds of the same class was also observed in the case of flavonols and phenolic acids. Furthermore, based on these measurements, no clear correlation between encapsulation efficiency and relative hydrophilicity of the compounds was observed. These observations suggest that the partitioning of compounds in cells from a juice matrix significantly depends on the interactions among the polyphenolic compounds and the composition of cells. The characterization of these interactions is beyond the scope of this study, but these results suggest that it may be possible to select cellular compositions among the diverse class of microbes that may promote the binding of selective polyphenols from a given plant extract and juice.

### 2.4. Stabilization of Bioactive Compounds against Heat Treatment

One of the important functionalities of encapsulation carriers is to protect the bioactive compounds from adverse environmental factors and food processing conditions. These adverse conditions may cause damage by oxidation, less favorable pH, and thermal induced reactions in bioactive compounds in food matrices. In order to produce a shelf-stable and microbially safe food, thermal processing methods such as pasteurization or sterilization are commonly used. Thus, in this study the effectiveness of the selected bacterial carrier in protecting and stabilizing the encapsulated juice polyphenols was evaluated. To evaluate the thermal stability of encapsulated bioactives, the cells were heat-treated at 90 °C for 1, 2, 5, 10, 20, 40, and 60 min in a temperature-controlled water bath. The heating conditions were selected based on prior studies [9,61]. After the heat treatment, the cell-encapsulated polyphenolics were extracted using the methods described in Section 3.10 of the material and methods section. The total antioxidant concentration of the extract was then measured using the FRAP assay. The control group of cells with encapsulated compounds but without the heat treatment were also extracted using the same approach and used for calculating the retention ratio during the treatment. The results in Figure 4 illustrate the percentage of total antioxidant capacity retained at each time point during the heating process. As observed in Figure 4, the bacterial carrier effectively protected the encapsulated compounds during thermal treatment. Approximately 93% of the antioxidant capacity for the encapsulated MG juice was retained after 1 h of heat treatment (Figure 4), whereas only 74% of the initial antioxidants were preserved without using encapsulation after the heat treatment of juice for 1 h. These observations indicated that cell carriers can effectively protect encapsulated antioxidant compounds against thermal stress.

In addition to the total antioxidant capacity, the retention of anthocyanins was also monitored during the heating process at 90 °C. The cells encapsulated with polyphenols from the juice matrix were sampled at 1, 2, 5, 10, 20, 40, and 60 min, and compared to the non-heated polyphenols encapsulated in cells from the juice. As described previously, the anthocyanin content retained in the cells was extracted using methanol and measured using a UV-Vis spectrometer. Figure 5 shows similar patterns of enhanced stability of anthocyanin compounds on cell carriers similar to the results in Figure 4. Despite the fact that the MG juice contains more colored pigments (Figure 2), these compounds seemed to be more susceptible to heat. Only 61% of the anthocyanin pigments in the MG juice were retained after 60 min of heat treatment. In contrast, 90% of the encapsulated anthocyanin pigments were preserved in the cell carriers. These results demonstrated that the bacterial cell carriers effectively protected encapsulated anthocyanins from degradation caused by the thermal treatment.

Overall, the results demonstrated that, after 60 min of heat treatment at 90 °C, more than 87% of the total antioxidant capacity and 90% of the anthocyanin content were recovered from the encapsulated MG as compared to the respective juice without encapsulation. The degradation of juice phenolics content including anthocyanin from heating were comparable with previous studies (between 15 and 30%) [9,61].

The thermal stability of the encapsulated active compounds was significantly higher than non-encapsulated MG juice. This protective effect of microcarriers has been observed in a range of encapsulation systems such as spray-drying particles [65] and emulsions [66]. However, comparable or higher percentages of antioxidant capacity and pigment content retention were observed using the cell carriers compared to the synthetic encapsulation carriers. For instance, more than 20% losses were observed for anthocyanins encapsulated in polymer matrices such as maltodextrin, mixture of maltodextrin and gum arabic, and soluble starch after treatment at 98 °C for 30 min [67], whereas cell carriers reduced the loss to less than 10% after the heating of the encapsulated samples at 90 °C for 60 min. The advantage of the cell carriers might be attributed to both the physical cellular structure and its complex chemical composition. As shown in Figure 1, the cell structures persisted through the encapsulation process, and literature has shown that some of the *Lactobacillus* strains can maintain structural integrity at elevated temperatures around 100–120 °C, for 30 to 60 min [68]. The robust structure is essential for protecting encapsulated bioactives, whereas colloidal encapsulation systems tend to destabilize both physically and chemically during encapsulation or in adverse environmental conditions [69]. Besides the physical structure, the antioxidant property of intracellular content of *L. casei* has also been reported [70]. Aguilar-Toalá et al. suggested that glutathione and other intracellular lipid and protein components might be involved in the antioxidant activities, which might in turn help stabilize and protect bioactive compounds encapsulated within the cell carrier. Therefore, cell carriers are an efficient encapsulation material for preserving the bioactive functions of the extracted polyphenolics during thermal processing.

In addition, encapsulation using cell carriers exhibits certain advantages in terms of the manufacturing process. In this study, we used *L. casei* cells to encapsulate a composite profile of polyphenolics with a basic temperature-controlled incubation. Currently, spray drying and freeze drying are the most commonly applied industrial techniques for microencapsulation and stabilization of plant polyphenolics from natural sources. Spray drying is a unit operation where liquid is atomized in a hot gas current to obtain a powder [71]. Freeze drying is an alternative drying method but less utilized due to the higher operational cost. While spray drying is prevalent with low cost, its limitations have also been extensively discussed. We observed 4 to 5 times higher amounts of anthocyanin content encapsulated in the cell carrier in this study when compared to spray-dried powder [72]. Despite variations in the raw material, loss of heat in sensitive compounds during spray drying might be due to the exposure to oxygen [73] and the thermal treatment (with typical inlet air temperature around 150–220 °C albeit with a short contact time [71,73]). In addition, the drying process may cause the loss of dried material due to wall deposition, low thermal efficiency [74], broad size distribution, and irregular microstructures [75]. Encapsulation using the preformed cellular structure of probiotic bacteria and passive incubation, on the other hand, significantly simplified the process with more uniform cellular size and microcellular structure.

## 3. Material and Method

### 3.1. Plant Material

Muscadine grapes were procured from the Coca-Cola Company, Atlanta, GA, USA. Cultivar is a predominant variety in the United States. Fruit was stored at 4 °C until juice processing.

### 3.2. Reagents and Standards

Phenolic standards (gallic acid (≥98%), (+)-catechin (≥99%), (-)-epicatechin (≥98%), caffeic acid (≥98%), coutaric acid (≥90%), quercetin (≥95%), and myricetin glycosides (≥99%)) were obtained from Sigma Chemical Co. (St. Louis, MO, USA). Keracyanin chloride (≥98%) were obtained from Sigma Chemical Co. as the anthocyanin standard. For HPLC analysis, phosphoric acid, 85 wt.% in H_2_O were obtained from Sigma Chemical Co. TPTZ (2,4,6-tripyridyl-s-triazine), and FeCl_3_·6H_2_O used for the FRAP assay were obtained from Thermo Fisher Scientific Inc. (Waltham, MA, USA). All solvents used in this study were HPLC grade.

### 3.3. Juice Processing

A slow masticating juicer (KOIOS, Model SHA1066, (Lodi, CA, USA) was used to prepare the fresh fruit juice from grapes. The fruits were rinsed with a deionized water and air dried before juice processing. Juice and pomace samples were collected separately from the juicer. Mass of fruit and produced juice were recorded, and the juice yield was calculated as a ratio of the mass of juice to the total fruit mass. Fresh juice was then divided into 5 mL aliquots and freeze-dried overnight. The dried juice powder was then reconstituted with 5 mL of DI water and centrifuged to remove any insoluble plant material.

### 3.4. Bacterial Strains and Cell Preparation

*Lactobacillus casei* (ATCC 393) were selected as a model for human probiotic bacteria. The stock strain is stored in a liquid nitrogen. M.R.S. agar and broth were used to culture this strain according to the ATCC protocol. Before the experiments, the stock strain was streaked onto M.R.S. agar plates and incubated overnight at 37 °C. A single colony from the agar plate was then used to inoculate liquid medium and incubated at 37 °C without agitation to achieve the stationary phase of bacterial culture. After centrifugal separation, the bacteria were inactivated using 70% ethanol for 30 min and washed with sterile Phosphate Buffered Saline (PBS). The bacteria were then suspended in sterile PBS at a concentration of approximately 10^10^ CFU/mL.

### 3.5. Encapsulation in Inactivated Cell Carriers

Bacteria pellet was collected after centrifugation (11,000 rpm for 5 min) and mixed with the reconstituted muscadine grape juice. The encapsulation process was carried out in a 4 °C cold room for 24 h with mild agitation. Mild agitation was performed by shaking tubes horizontally at 100 rpm. After encapsulation, the aqueous juice matrix and cell carriers were separated by centrifugation (11,000 rpm for 5 min) and collected separately. The cells were washed once using sterile PBS buffer. Supernatant after wash was also collected to analyze the non-infused bioactive compounds using the FRAP assay described in Section 3.6.

### 3.6. Total Antioxidant Capacity Measurements

Total antioxidant capacity of juice matrix before and after encapsulation was quantified using the Ferric Reducing Antioxidant Power (FRAP) assay. Antioxidant activity was selected as a representation of the total bioactive compound concentration in the juice [76]. The changes in the antioxidant content of the juice after encapsulation was evaluated to assess the encapsulation efficiency of diverse class of bioactive compounds. The protocol for measuring FRAP activity was adapted from Benzie and Strain (1996) [77]. The stock solutions included 300 mM acetate buffer (pH 3.6), 10 mM TPTZ solution in 40 mM HCl, and 20 mM FeCl_3_·6H_2_O solution. The fresh working solution was prepared by mixing 25 mL acetate buffer, 2.5 mL TPTZ solution, and 2.5 mL FeCl_3_·6H_2_O solution and then warmed at 37 °C before using. Fruit juice before and after encapsulation (150 μL) was allowed to react with 2850 μL of the FRAP solution for 30 min in the dark condition. Change in color of the solution (ferrous tripyridyltriazine complex) was quantified using a UV-Vis measurement at 593 nm using a spectrometer. The standard curve was generated using a range of Trolox solutions between 25 and 800 μM. Results were expressed in μM T.E./mL fresh juice. The samples were diluted in case the absorbance value measured for the samples was over the linear range of the standard curve.

### 3.7. Colored Pigment Measurements

Anthocyanin content in juice before and after encapsulation was also measured using a UV-Vis spectrometry. Grape juice is a significant source of plant anthocyanins and changes in the level of anthocyanins in a juice matrix before and after encapsulation also represent a measure of encapsulation of water-soluble pigments in bacterial cells. The absorbance value of the clarified samples was scanned from 250 nm to 600 nm, and a peak intensity was recorded at 530 nm for all the samples [78]. The samples were diluted accordingly to avoid saturation in the absorbance signal. Standard curves were constructed using different concentrations of keracyanin chloride and anthocyanin content was represented as keracyanin equivalent content. 

### 3.8. Confocal Fluorescence Imaging

Confocal Laser Scanning Microscopy (CLSM) images of bacterial cells after encapsulation with and without incubation with muscadine juice sample were collected using a Zeiss LSM 510 upright microscope (Carl Zeiss AG, Jena, Germany) with 40×/1.1 water objective. Each sample was excited at 405 nm using an argon diode laser. Emission (xyz) scans were acquired using a 500–550 nm bandpass emission filter. Lambda (xyλ) scans of each sample were collected over a range of 470–670 nm with 20 nm step size. The average intensity of the images acquired at different wavelengths during the lambda scan was measured using ImageJ software and plotted using an Origin 8.0 (Origin lab, Northampton, MA, USA).

### 3.9. Phenolic Extraction and HPLC-DAD Analysis

Phenolic compounds were extracted by mixing 2 mL of the reconstituted MG juice sample with 13 mL of acidified methanol (with 1% HCl). After mixing using a vortexer, the mixture was sonicated using a bath sonicator for 10 min and the extract was separated from the remaining juice solids by centrifugation at 5500 rpm for 5 min. The samples were then diluted 10-fold with milliQ water for HPLC-DAD analysis. To assess encapsulation efficiency and yield in cell-based carriers, phenolic content in the aqueous phase before and after encapsulation process was quantified.

Chromatography separation and detection of phenolic compounds were performed on an Agilent 1260 Infinity reverse phase HPLC (RP-HPLC)-D.A.D. system (Santa Clara, CA, USA) equipped with a thermostatic autosampler, thermostatic column compartment, and a diode array detector [15] according to a method adapted from Plaza et al. [79]. An Agilent PLRP-S 100 Å (4.6 × 150 mm, 3 μm) column with an Agilent 3 × 5 mm guard column (PL1310-0016; Santa Clara, CA, USA) was used at a temperature of 35 °C for all the analysis. Mobile phase A: 1.5% phosphoric acid solution. Mobile phase B: acetonitrile solution containing 20% (*v*/*v*) mobile phase A. The gradient protocol for HPLC separation and analysis was as follows: 0 min, 94% solvent A; 73 min, 69% A; 78 min, 38% A; and 90 min, 94% A. The flow rate was 1 mL/min and the injection volume for all samples was 10 μL. Samples were filtered through 0.45 μm type H.A. Millipore filters (Millipore Corp., Bedford, MA, USA) prior to injection.

Absorbance spectra were recorded from 250 nm to 600 nm. The eluted compounds were monitored and identified based on spectral and retention time comparisons with standards at multiple wavelengths, including 280 nm for Flavanol [gallic acid, (+)-catechin and (−)-epicatechin] and polymeric phenols, 320 nm for hydroxycinnamates (caffeic acid and coutaric acid), and 360 nm for flavonol and derivatives (quercetin and myricetin 3-O-glucosides), respectively, using the D.A.D. detector. External calibration curves were constructed for gallic acid, (+)-catechin, (−)-epicatechin, caffeic acid, coutaric acid, quercetin, and myricetin glycosides were used for quantification of the target compounds. Polymeric phenols were quantified as catechin equivalents. Chromatograms were integrated using the Agilent CDSChemStation Software (Agilent, Santa Clara, CA, USA).

### 3.10. Heat Treatment and Extraction

The thermal stabilities of the encapsulated bioactive compounds were evaluated using a thermostatic water bath at 90 °C for up to 60 min. A 1 mL suspension of the cells with encapsulated compounds and 1 mL of juice alone were added to the prewarmed 20 mL glass vials (Thermo Scientific™ B780020, Waltham, MA, USA) and incubated in the dark for 1, 2, 5, 10, 20, 40, and 60 min. The concentration of the total antioxidant contents in the juice sample and the cell encapsulated sample were maintained the same. After the treatment, 1 mL acidified methanol was added to each vial. Bead-beating at 6.0 m/s for 30 s for 3 times (FastPrep-24™ 5G Instrument, MP Biomedicals, Irvine, CA, USA) was then carried out to facilitate thorough extraction. Finally, the homogenized samples were sonicated using a bath sonication device (Branson 2510 Ultrasonic Cleaner, Branson Ultrasonics Corp., Danbury, CT, USA) for 10 min. The methanolic extract was then centrifuged to remove cell debris and the supernatant was used for subsequent anthocyanin measurement and total antioxidant capacity quantification as described in Section 3.6 and Section 3.7

### 3.11. Statistical Analysis

Statistical analysis was performed using the GraphPad Prism software V.7.0a (Graphpad Software, Inc., La Jolla, CA, USA). All experiments were performed in triplicates, three independent sample measurements and treatments. The significant differences between the treatments were determined through one-way ANOVA with a significance level at *p* < 0.05. Multiple comparison was then carried out using the Holm-Šídák test with a significance level at 0.05.

## 4. Conclusions

In summary, microencapsulation leveraging inactivated probiotic cells (*L. casei*) as the pre-formed microcarrier could separate and enhance the stability of antioxidant compounds of composite profiles from crude fruit juice material. As identified in this study, the profiles include anthocyanins and a variety of phenolic compounds. Compared to the original MG fruit juice used in this study, the cell carriers contained more than 60% of the antioxidant capacity and anthocyanin content using the simple incubation protocol. After heat treatment at 90 °C for 60 min, more than 87% of the encapsulated antioxidant capacity and more than 90% of the anthocyanin content were preserved within the cell carriers. This approach presents an economic and scalable technique to better utilize waste and by-products of the food processing industry. Future studies could be carried out to assess the potential of encapsulating phytochemicals of other plant sources, modification of the cell carrier structures and encapsulation protocol to further enhance the encapsulation efficiency, and any additional probiotic benefits that the bacterial carrier might have.

## Figures and Tables

**Figure 1 molecules-27-05821-f001:**
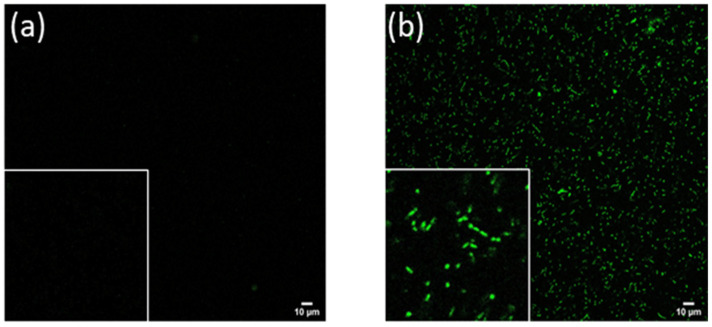
Confocal fluorescence microscopy of juice encapsulated cells with a 405 nm excitation, and 500–550 nm emission; (**a**) *L. casei* control cells; (**b**) *L. casei* cells encapsulated with MG juice. Lower left is the zoomed-in view of the same image: zoomed-in views in (**b**) demonstrated the uniform intracellular localization of polyphenolic compounds in the cell carriers.

**Figure 2 molecules-27-05821-f002:**
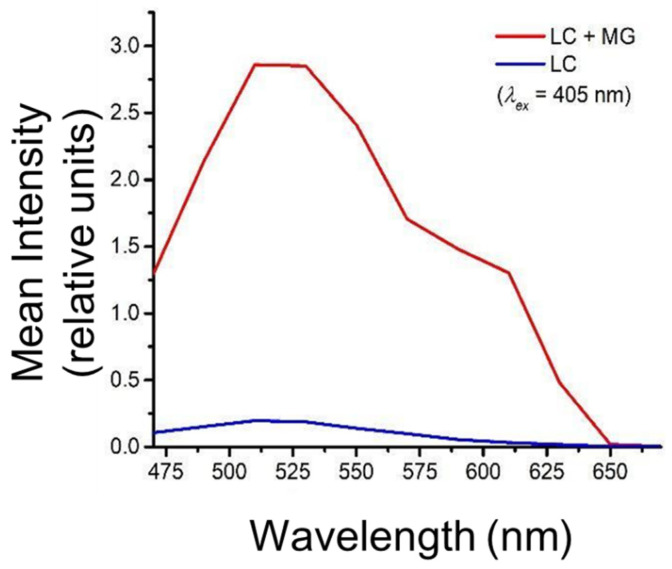
Mean intensity of confocal lambda-scans over the range of 470–670 nm with a 405 nm excitation.

**Figure 3 molecules-27-05821-f003:**
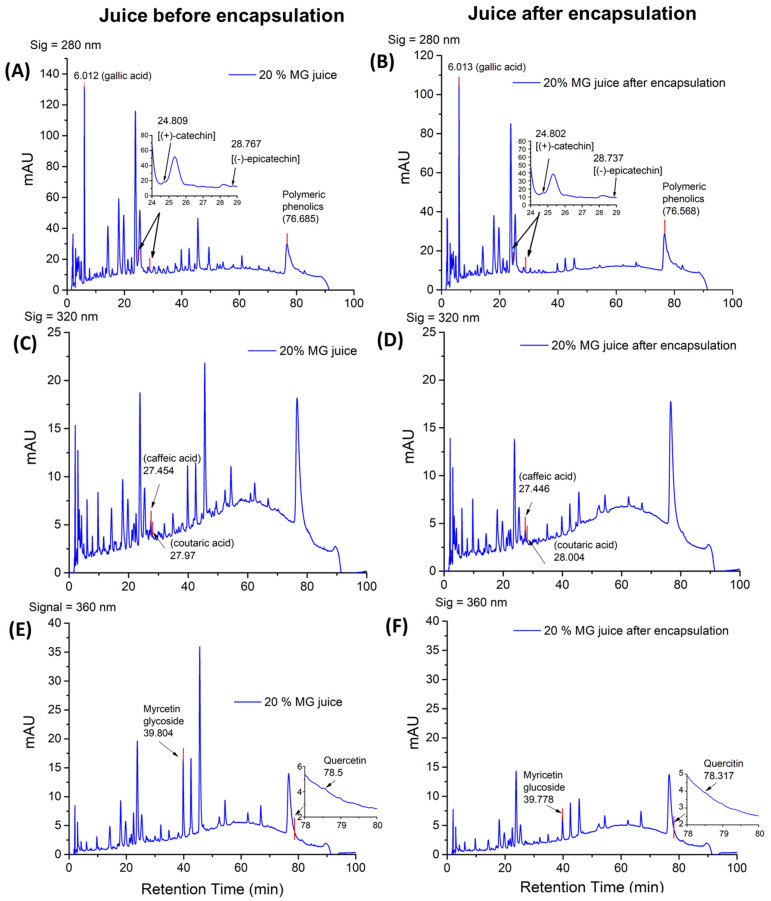
Chromatograms corresponding to absorbance peaks at 280 nm (**A**,**B**), 320 nm (**C**,**D**), and 360 nm (**E**,**F**) with assigned peaks for different phenolic compounds from 20% MG juice before (left) and after (right) encapsulation. Insets in Figure (**A**,**B**) illustrate a zoomed-in chromatogram region between a retention time of 24 to 29 min on the HPLC column to highlight the assigned peak for catechin and epicatechin in juice before and after encapsulation, respectively. Insets of (**E**,**F**) ill zoomed-in chromatogram section between the retention time of 78 to 80 min on the HPLC column to highlight the assigned peak for quercetin in juice before and after encapsulation.

**Figure 4 molecules-27-05821-f004:**
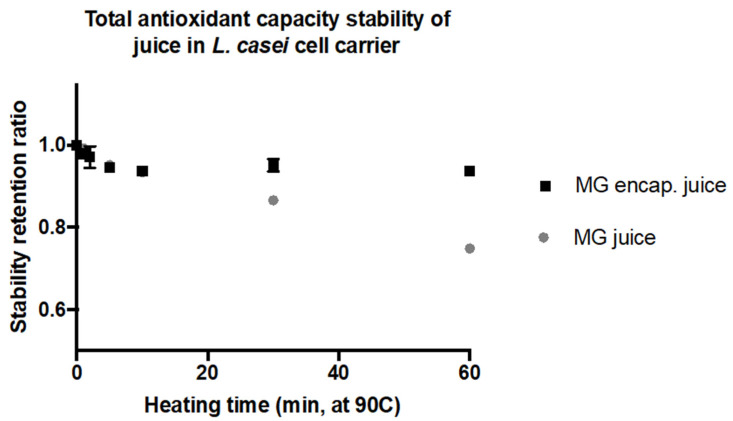
Retention of antioxidant content of phenolic bioactives from grape juice upon selected heat treatment both with and without encapsulation in the selected cell-carrier, after treatment juice samples were extracted with 85% methanol and quantified using the FRAP assay.

**Figure 5 molecules-27-05821-f005:**
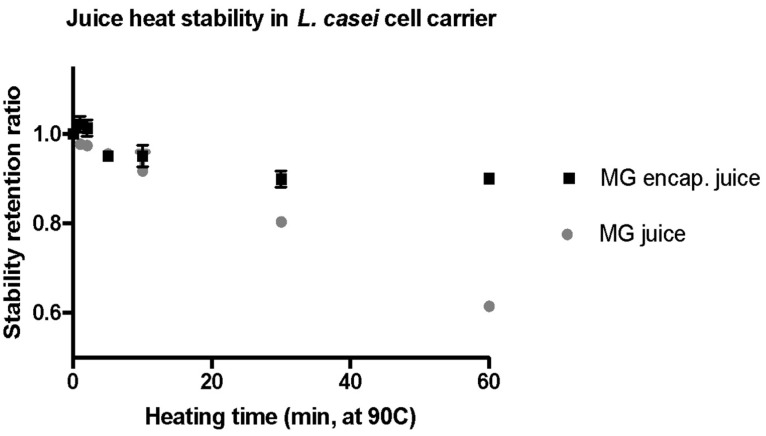
Stability of juice pigments with and without encapsulation under heat treatment. After treatment, the MG juice was extracted with 85% methanol and quantified using UV-Vis measurements at 530 nm.

**Table 1 molecules-27-05821-t001:** Characterization of MG juice matrix and the encapsulation efficiency of the polyphenolics from the juice using *L. casei* cell carrier. For this characterization, the antioxidant activity of the juice before and after encapsulation was quantified.

	MG Juice Matrix
Juice matrix anthocyanin (λmax = 530 nm)	8.21 (±0.07) μM/mL
Encapsulation efficiency of anthocyanin content	66.97% (±1.68%)
Total antioxidant capacity (FRAP assay, T.E. µM/mL)	3436.43 (±40.41)
Total antioxidant capacity retention in cell carriers after encapsulation	72.67% (±0.70%)

**Table 2 molecules-27-05821-t002:** Phenolic profile of MG juice encapsulated compounds in *L. casei* cell carrier.

Analyzed Polyphenols	MG
Class	Compound	Molecular Structure	Molecular Weight (g/mol)	Log *p* Value	Content in 20% Matrix (mg/mL)	Juice Encapsulation Efficiency	Statistical Significance (*p* < 0.05)
Flavanol	(+)-Catechin	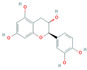 (C_15_H_14_O_6_)	290.27	0.41(Poaty, 2009)	16.78	17.40% (±14.12%)	-
(−)-Epicatechin	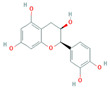 (C_15_H_14_O_6_)	290.27	1.8	3.18	18.77% (±9.08%)	*
Phenolic acid	Gallic Acid	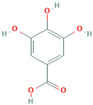 (C_7_H_6_O_5_)	170.12	0.7	1.69	18.43% (±1.95%)	*
Caffeic Acid	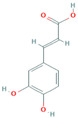 (C_9_H_8_O_4_)	180.16	1.15	0	-	-
Coutaric Acid	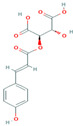 (C_13_H_12_O_8_)	296.23	−1.32(Jana, 2017)	0.07	10.24% (±6.06%)	-
Flavonol and derivatives	Quercetin	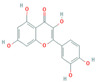 (C_15_H_10_O_7_)	302.23	1.48	21.70	2.83%(±3.75%)	
Myricetin 3-glucoside	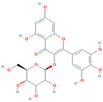 (C_21_H_20_O_13_)	480.40	−0.45	3.53	69.85% (±3.03%)	*
Polymeric phenols	Polymeric phenols	-(a mixture of polymeric phenols)	-	-	26.09	97.97% (±2.53%)	*

## Data Availability

The data is available upon request to the corresponding author.

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
