# Peer review of "Lactic Acid Bacteria Simultaneously Encapsulate Diverse Bioactive Compounds from a Fruit Extract and Enhance Thermal Stability"

_molecules, 2022, doi:10.3390/molecules27185821_

Round 1

Reviewer 1 Report

The work focuses on the ability of inactivated probiotic bacteria to encapsulate phytochemicals present in juice, which increases the stability of the encapsulated compounds against thermal degradation.

The work is interesting and raises several questions. There are though some comments about methodology that need to be adressed prior to publication.

On page 3, line 127 and on... the method for determination the extent antioxidant capacity is presented, FRAP. There is no description though of what FRAP means. Please correct.

The structure of anthocyanin should be shown. 

As described, the cells are separated by centrifugation and washed with water. Was the content of phytochemicals/antioxidant capacity meaured as well on this wash water? This should be accounted as non-bound compounds.

Table 1,. the value 8.21, juice matrix anthocyanin. What does it refer to? Absorbance? In the text, page 4 line 160 there is a mention of a concentration that does not match that number. Please revise and correct.

The legends in Figure 2 should have an explanatory for LC. The scans for the juice alone should be included. Is there a shift in the spectra due to encapsulation? A shift might help explain how the molecules are encapsulated.

LC is a gram positive bacteria with a thick extracellular wall, quite impenetrable. How come polyphenolic compounds are enriched inside the cell? How are these penetrating the cell? The wall is believed to protect the cell against large molecules leaving only small ones to diffuse through and translocate into the cell. Could it be that the polyphenolic compounds are bound only in the cell wall?

Regarding Fig 4 and 5. How was the measuremnets done? Is it done on the juiice and then the cells are separated by centrifugation, washed and subjected to the heat treatment? Or was the separation done before the heat treatment? How was the measurements of the juice done?  The methods section should be clarified.  

Are these triplicates of the same sample, or are these three different samples arising from 3 different juices + LC mixtures?  Or is this replicates of the same juice + LC mixtures? Replicates on different juices + LC should be done to strengthen the results.

If the encapsulated molecules are water soluble, encapsulation shouldn't have an effect upon the stability against heating. Clearly the molecules are bound to something. The authors mention previous literature about binding with proteins and lipids. What is the most likely scenario?

Author Response

Dear Editor, Thanks for organizing the review of our manuscript. We have carefully reviewed the feedback from the reviewers and have revised our manuscript. We have also prepared a detailed response to the reviewers' comments as described below.

Regards,

Nitin

Professor, Departments of Food Science and Technology and Biological and Ag. Engineering

University of California, Davis

Response to the Reviewers' Feedback

The work focuses on the ability of inactivated probiotic bacteria to encapsulate phytochemicals present in juice, which increases the stability of the encapsulated compounds against thermal degradation. The work is interesting and raises several questions. There are though some comments about the methodology that need to be addressed prior to publication.

Response: Thanks for your valuable feedback. We have addressed your comments in the section below

On page 3, line 127, and on... the method for determining the extent antioxidant capacity is presented, FRAP. There is no description though of what FRAP means. Please correct.

Response: We have added a description of FRAP (Ferric reducing antioxidant power) and also added a statement referring to the method section where the details of the method (section 3.6) and references are listed (ref no. 41, 76, and 77).  

  1. Merola, E. T.; Catherman, A. D.;  Yehl, J. B.; Strein, T. G., Determination of total antioxidant capacity of commercial beverage samples by capillary electrophoresis via inline reaction with 2,6-dichlorophenolindophenol. J Agric Food Chem 2009, 57 (15), 6518-23.
  2. Sharayei, P.; Azarpazhooh, E.; Ramaswamy, H. S., Effect of microencapsulation on antioxidant and antifungal properties of aqueous extract of pomegranate peel. J Food Sci Technol 2020, 57 (2), 723-733.
  3. Estévez, L.; Queizán, M.;  Mosquera, R. A.;  Guidi, L.;  Lo Piccolo, E.; Landi, M., First Characterization of the Formation of Anthocyanin–Ge and Anthocyanin–B Complexes through UV–Vis Spectroscopy and Density Functional Theory Quantum Chemical Calculations. Journal of Agricultural and Food Chemistry 2021, 69 (4), 1272-1282.

The structure of anthocyanin should be shown. 

Response: Using our spectroscopic measurements, anthocyanins are measured as a family of compounds not a specific compound within this broader family and hence the structure of an individual compound may not be relevant. However, we have added a reference (ref no 42) that illustrates the structure and properties of this class of compounds.  

Reference 42. Yuzuak, S.; Xie, D.-Y., Anthocyanins from muscadine (Vitis rotundifolia) grape fruit. Current Plant Biology 2022, 30, 100243.

As described, the cells are separated by centrifugation and washed with water. Was the content of phytochemicals/antioxidant capacity meaured as well on this wash water? This should be accounted as non-bound compounds.

Response: Yes, we did combine the wash water measurements with the residual juice samples to account for any non-bound compounds that could be removed by the washing process. We have revised the method section to further clarify this (Lines 424 and 425).

Table 1,. the value 8.21, juice matrix anthocyanin. What does it refer to? Absorbance? In the text, page 4 line 160 there is a mention of a concentration that does not match that number. Please revise and correct.

Response: We have revised the reported value and its associated unit in Table 1 to match the number in the text. This value represents the concentration value measured based on the absorbance measurement and the calibration curve.

The legends in Figure 2 should have an explanatory for LC. The scans for the juice alone should be included. Is there a shift in the spectra due to encapsulation? A shift might help explain how the molecules are encapsulated.

Response: The LC spectra are shown in Fig. 3. We have revised the figure to clarify that it includes the scans both before (20 % juice) and after encapsulation (20% juice after encapsulation) as suggested by the reviewer. In addition, the legends on the figure have been revised.

LC is a gram positive bacteria with a thick extracellular wall, quite impenetrable. How come polyphenolic compounds are enriched inside the cell? How are these penetrating the cell? The wall is believed to protect the cell against large molecules leaving only small ones to diffuse through and translocate into the cell. Could it be that the polyphenolic compounds are bound only in the cell wall?

Response: To evaluate the localization of polyphenolic compounds, confocal fluorescence microscopy data were acquired, as shown in Fig. 1. The results indicate that the compounds are localized uniformly in the cell matrix, and there is no significant background signals. These results suggest that the compound penetration in the cells is not limited to the cell wall only but includes intra-cellular distribution. In this research, the inactivation of probiotic cells with heating may be one of the factors influencing the enhancement in permeability. Further research studies will evaluate the role of various factors influencing the partitioning properties and permeability of compounds in cellular matrixes.

Regarding Fig 4 and 5. How was the measuremnets done? Is it done on the juiice and then the cells are separated by centrifugation, washed and subjected to the heat treatment? Or was the separation done before the heat treatment? How was the measurements of the juice done?  The methods section should be clarified.  

Response: These measurements were done in two independent sets of experiments. The juice samples were independently heated from the cells with encapsulated compounds from the juice. Thus, no separation is required between the juice and the cells. We have revised the method section 3.10 (lines 488 – 492) to clarify this.  

Are these triplicates of the same sample, or are these three different samples arising from 3 different juices + LC mixtures?  Or is this replicates of the same juice + LC mixtures? Replicates on different juices + LC should be done to strengthen the results.

Response: All measurements are three independent sample measurements and treatments. We have clarified this in the methods section 3.11 (lines 501 and 502)

If the encapsulated molecules are water soluble, encapsulation shouldn't have an effect upon the stability against heating. Clearly the molecules are bound to something. The authors mention previous literature about binding with proteins and lipids. What is the most likely scenario?

Response: We appreciate the suggestions and thoughts. We agree with the reviewer that both binding and compartmentalization of the molecules in the cellular environment is one of the key factors influencing their stability. It is likely that molecules with higher hydrophobicity may associate with lipid-rich environments in cells, while molecules such as phenolic acids, and flavanols may associate with proteins in cells.

Reviewer 2 Report

The Manuscript submitted by Dou and coworkers explores the use of Lactic acid bacteria to encapsulate multiple bioactives from a fruit juice water extract. The enhanced thermal stability during processing is also analyzed. The manuscript is well written and organized. Results and discussion sections are well supported by data displayed in graphs and tables. The topic is of interest and fits the general scope of the journal.

Author Response

The Manuscript submitted by Dou and coworkers explores the use of Lactic acid bacteria to encapsulate multiple bioactives from a fruit juice water extract. The enhanced thermal stability during processing is also analyzed. The manuscript is well written and organized. Results and discussion sections are well supported by data displayed in graphs and tables. The topic is of interest and fits the general scope of the journal.

Response: Thanks for your encouraging feedback.

Reviewer 3 Report

This study developed an innovative cell-based carrier that can simultaneously encapsulate multiple phytochemicals from a complex plant source. The quantification of total antioxidant capacity of juice matrix before and after encapsulation by the FRAP assay, the measurement of anthocyanin content in the juice before and after encapsulation by UV-Vis spetrophotometry, the observation of intracellular distribution of encapsulated compounds by Confocal Laser Scanning Microscopy, and the quantification of phenolic compounds before and after incubation with cells by HPLC proved that the cell carrier had good encapsulation efficiency and performance of various phytochemicals. 

  Considering the well-done experiments and inspiring results to this cell-based carrier for multiple phytochemicals encapsulation, this manus

cript is worthy of publication before some minor revisions:

1.The picture of molecular structure of catechins, epicatechin, gallic

acid, quercetin, and myricetin 3-glucoside in Table 2 are not complete. Please replace it with a complete picture of structure.

2.Table 2 shows that the juice encapsulation efficiency of catechin and epicatechin are 17.40%(±14.12%) and 18.77%(±9.08%), respectively

The results are too unstable to prove the stability of the encapsulation system.

3.In Figure 3, the peak assignment of catechin, epicatechin and quercetin are not clear, please correct it.

4.In order to demonstrate the protective effect of cellular carriers on encapsulated bioactive compounds, only environmental factors related to heat were studied. Some other relevant environmental factors should also be considered.

5.Please add another parenthesis between the third and fourth lines on page 12.

Author Response

Reviewer 3

This study developed an innovative cell-based carrier that can simultaneously encapsulate multiple phytochemicals from a complex plant source. The quantification of total antioxidant capacity of juice matrix before and after encapsulation by the FRAP assay, the measurement of anthocyanin content in the juice before and after encapsulation by UV-Vis spetrophotometry, the observation of intracellular distribution of encapsulated compounds by Confocal Laser Scanning Microscopy, and the quantification of phenolic compounds before and after incubation with cells by HPLC proved that the cell carrier had good encapsulation efficiency and performance of various phytochemicals. 

  Considering the well-done experiments and inspiring results to this cell-based carrier for multiple phytochemicals encapsulation, this manuscript is worthy of publication before some minor revisions

Response: We appreciate the feedback. Thanks for your encouraging comments.

1.The picture of molecular structure of catechins, epicatechin, gallic

acid, quercetin, and myricetin 3-glucoside in Table 2 are not complete. Please replace it with a complete picture of structure.

Response: Sorry for this oversight. We have revised the Table 2 to ensure that the figures are not covered by the text in the table.  

2.Table 2 shows that the juice encapsulation efficiency of catechin and epicatechin are 17.40%(±14.12%) and 18.77%(±9.08%), respectively The results are too unstable to prove the stability of the encapsulation system.

Response: We appreciate this feedback. The absorbance of catechin and epi-catechin was measured at 280 nm. At this wavelength, the juice matrix has a large peak that is adjacent to the peaks for catechin and epi-catechin. In the inset of Figures 3A and 3B, we have shown the zoomed-in chromatogram between the retention time of 24 – 29 min on an HPLC column to highlight the assigned peak for catechin and epicatechin in juice before and after encapsulation. The accuracy in quantification for these peaks of the catechin and epicatechin is limited as the intensity of these peaks is peak is low and is next to a larger peak. In addition, the baseline of the spectra was not flat in this wavelength range, resulting in some noise in the measurement. We believe analytical methods such as HPLC-MS may address some of these limitations. We will definitely consider these approaches for our future studies. 

3.In Figure 3, the peak assignment of catechin, epicatechin and quercetin are not clear, please correct it.

Response: We have added zoomed-in sections in the revised Figs. 3A and 3B to clarify the peak assignments.

4.In order to demonstrate the protective effect of cellular carriers on encapsulated bioactive compounds, only environmental factors related to heat were studied. Some other relevant environmental factors should also be considered.

Response: We appreciate the recommendation. We think oxidative and pH stability could be considered. We have revised our discussion to indicate the possibilities of these other environmental factors in influencing stability (Section 2.4, lines 307 – 312). We will consider them in our future studies.  

5.Please add another parenthesis between the third and fourth lines on page 12.

Response: We have corrected for the typographical error in the revised manuscript.  

1 Should be described in more detail in section 3.5 --- Encapsulation in inactivated cell carriers. The encapsulation process was carried out in a 4°C cold room for 24 hours with mild agitation. How to set mild agitation intensity?

Response: We have added the details on the mild agitation process in the methods section 3.5 (lines 421 – 422). Agitation was mainly designed to prevent cells from settling during extended incubation.

2 How reproducible and reproducible is this study?

Response: The results are highly reproducible, and all the experiments were conducted in triplicates (three independent repeats).

3 The title of the manuscript is too long.

Response: Thanks for the suggestion. We have revised the title- Lactic acid bacteria simultaneously encapsulates diverse bioactive compounds from a fruit extract and enhances thermal stability  

4 Please quote this paper.

Phenolic-Rich Plant Extracts with Antimicrobial Activity: An Alternative to Food Preservatives and Biocides?

Oulahal N, Degraeve P.Front Microbiol. 2022 Jan 4;12:753518. doi: 10.3389/fmicb.2021.753518. 2021.PMID: 35058892 Free PMC article. Review.

For instance, several authors recently suggested that natural phenolic-rich extracts could not only extend the shelf-life of foods by controlling bacterial contamination but could also coexist with probiotic lactic acid bacteria in food systems to provide enhanced health benefits to humans.

Response: We have added the reference (ref no 27) suggested by the reviewer in the introduction section and highlighted the potential of phenolic extract's co-existence with probiotic lactic acid bacteria.

Reference 27. Oulahal, N.; Degraeve, P. Phenolic-Rich Plant Extracts With Antimicrobial Activity: An Alternative to Food Preservatives and Biocides? Frontiers in microbiology, 2021, p. 753518.

Reviewer 4 Report

1 Should be described in more detail in section 3.5 --- Encapsulation in inactivated cell carriers. The encapsulation process was carried out in a 4°C cold room for 24 hours with mild agitation. How to set mild agitation intensity?
2 How reproducible and reproducible is this study?
3 The title of the manuscript is too long.
4 Please quote this paper.
Phenolic-Rich Plant Extracts With Antimicrobial Activity: An Alternative to Food Preservatives and Biocides?
Oulahal N, Degraeve P.Front Microbiol. 2022 Jan 4;12:753518. doi: 10.3389/fmicb.2021.753518. 2021.PMID: 35058892 Free PMC article. Review.
For instance, several authors recently suggested that natural phenolic-rich extracts could not only extend the shelf-life of foods by controlling bacterial contamination, but could also coexist with probiotic lactic acid bacteria in food systems to provide enhanced health benefits to human.

Author Response

1 Should be described in more detail in section 3.5 --- Encapsulation in inactivated cell carriers. The encapsulation process was carried out in a 4°C cold room for 24 hours with mild agitation. How to set mild agitation intensity?

Response: We have added the details on the mild agitation process in the methods section 3.5 (lines 421 – 422). Agitation was mainly designed to prevent cells from settling during extended incubation.

2 How reproducible and reproducible is this study?

Response: The results are highly reproducible, and all the experiments were conducted in triplicates (three independent repeats).

3 The title of the manuscript is too long.

Response: Thanks for the suggestion. We have revised the title- Lactic acid bacteria simultaneously encapsulates diverse bioactive compounds from a fruit extract and enhances thermal stability  

4 Please quote this paper.

Phenolic-Rich Plant Extracts with Antimicrobial Activity: An Alternative to Food Preservatives and Biocides?

Oulahal N, Degraeve P.Front Microbiol. 2022 Jan 4;12:753518. doi: 10.3389/fmicb.2021.753518. 2021.PMID: 35058892 Free PMC article. Review.

For instance, several authors recently suggested that natural phenolic-rich extracts could not only extend the shelf-life of foods by controlling bacterial contamination but could also coexist with probiotic lactic acid bacteria in food systems to provide enhanced health benefits to humans.

Response: We have added the reference (ref no 27) suggested by the reviewer in the introduction section and highlighted the potential of phenolic extract's co-existence with probiotic lactic acid bacteria.

Reference 27. Oulahal, N.; Degraeve, P. Phenolic-Rich Plant Extracts With Antimicrobial Activity: An Alternative to Food Preservatives and Biocides? Frontiers in microbiology, 2021, p. 753518.